# AurkA nuclear localization is promoted by TPX2 and counteracted by protein degradation

Italia Anna Asteriti[1],*, Federica Polverino[1],*, Venturina Stagni[1,2], Valentina Sterbini[1], Camilla Ascanelli[3], Francesco Davide Naso[1], Anna Mastrangelo[1], Alessandro Rosa[4,5], Alessandro Paiardini[6], Catherine Lindon[3], Giulia Guarguaglini[1]

The AurkA kinase is a well-known mitotic regulator, frequently overexpressed in tumors. The microtubule-binding protein TPX2 controls AurkA activity, localization, and stability in mitosis. Non-mitotic roles of AurkA are emerging, and increased nuclear localization in interphase has been correlated with AurkA oncogenic potential. Still, the mechanisms leading to AurkA nuclear accumulation are poorly explored. Here, we investigated these mechanisms under physiological or overexpression conditions. We observed that AurkA nuclear localization is influenced by the cell cycle phase and nuclear export, but not by its kinase activity. Importantly, AURKA overexpression is not sufficient to determine its accumulation in interphase nuclei, which is instead obtained when AURKA and TPX2 are co-overexpressed or, to a higher extent, when proteasome activity is impaired. Expression analyses show that AURKA, TPX2, and the import regulator CSE1L are co-overexpressed in tumors. Finally, using MCF10A mammospheres we show that TPX2 co-overexpression drives protumorigenic processes downstream of nuclear AurkA. We propose that AURKA/TPX2 co-overexpression in cancer represents a key determinant of AurkA nuclear oncogenic functions.

## Introduction

AurkA is a member of the Aurora family of serine/threonine kinases ([1], [2]). AurkA is a well-known mitotic regulator, with key roles in mitotic entry, centrosome maturation, and spindle organization through phosphorylation of several substrates, such as Cdk1 and Plk1 ([3], [4], [5], [6]). AurkA levels are cell cycle–regulated; they increase in the late S and G2 phases, peak in mitosis, and rapidly decrease at the mitotic exit when the kinase is degraded in a proteasome-dependent manner ([7], [8]). Interaction with its major activator TPX2 is required for complete AurkA activation; TPX2 is also a regulator of AurkA stability and localization at spindle microtubules ([9], [10], [11], [12]). AURKA overexpression is frequently observed in many cancer types ([13], [14]) and targeting the kinase is studied as an anti-cancer therapeutic approach ([13], [15], [16], [17]). Inhibitors of AurkA kinase activity are under evaluation in clinical trials, but moderate effects and partial specificity of action are observed ([14], [15], [16], [17], [18]). In recent years, evidence of AurkA non-mitotic roles, also in the G0/G1 phases, is emerging, for example, involvement in neurite outgrowth in post-mitotic neurons, primary cilium disassembly, DNA replication, and regulation of mitochondrial morphology and dynamics ([19], [20], [21]). We recently reviewed published evidence, integrated with data mining searches, correlating the oncogenic potential of AurkA with its increased localization at the interphase nucleus in tumors (solid and hematological) and hence with non-mitotic functions ([22]). In particular, in breast cancer AurkA nuclear localization has been proposed as a prognostic marker for poor survival ([23]) and has been shown to associate with transcriptional up-regulation ([24], [25]) and stabilization ([26]) of known oncogenes, such as FOXM1 and myc family members. A role of nuclear AurkA in activation of hypoxia transcriptional programs has been also recently reported in breast cancer, a condition that drives cell migration, morphological changes, and increased stemness, thus determining dissemination and metastases at other organs ([27]). Interestingly, some of the described roles of nuclear AurkA are reported as kinase-independent (reviewed in reference [22]). These observations strengthen the interest in nuclear roles of AurkA in cancer development and progression. Still, how AurkA nuclear localization is regulated under physiological and pathological conditions has been so far poorly investigated.

In the present study, we investigated the nuclear localization of AurkA to clarify the molecular mechanisms through which it is regulated. We analyzed non-mitotic AurkA localization in a

[1]Institute of Molecular Biology and Pathology, National Research Council of Italy, c/o Sapienza University of Rome, Rome, Italy    [2]Istituto di Ricovero e Cura a Carattere Scientifico, Fondazione Santa Lucia, Signal Transduction Unit, Rome, Italy    [3]Department of Pharmacology, University of Cambridge, Cambridge, UK    [4]Center for Life Nano- < Neuro-Science, Fondazione Istituto Italiano di Tecnologia, Rome, Italy    [5]Department of Biology and Biotechnologies "C. Darwin," Sapienza University of Rome, Rome, Italy    [6]Department of Biochemical Sciences, Sapienza University of Rome, Rome, Italy

Correspondence: giulia.guarguaglini@uniroma1.it; lia.asteriti@uniroma1.it
Francesco Davide Naso's present address is IRCCS Santa Lucia Foundation, Rome, Italy
*Italia Anna Asteriti and Federica Polverino contributed equally to this work

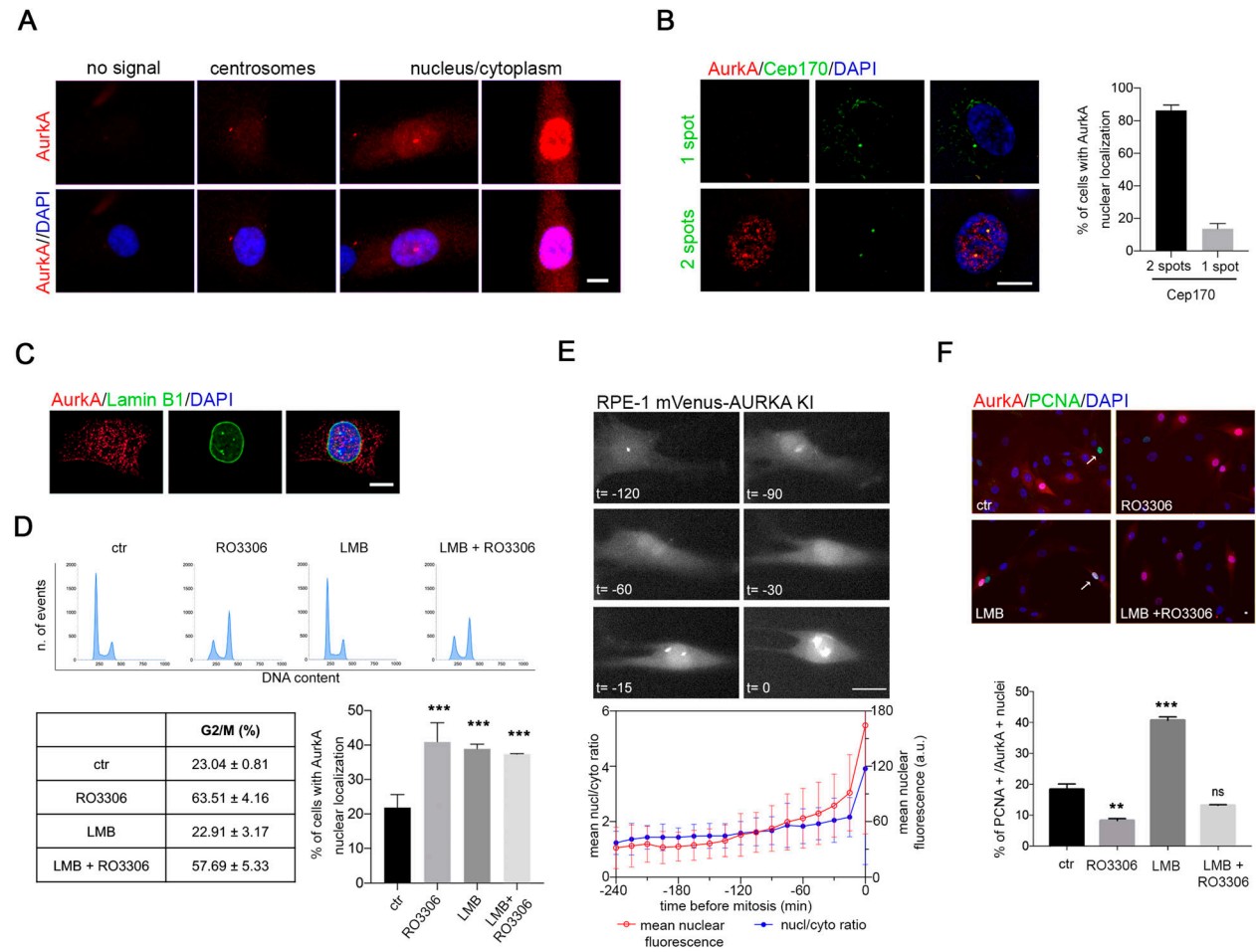

**Figure 1. Nuclear AurkA localization is correlated with the G2 phase.**
**(A)** IF panels show the interphasic AurkA localization patterns in hTERT RPE-1 cells. **(B)** Representative images of interphase cells with one (upper panels) or two (lower panels) spots of Cep170. Histograms represent the association of the nuclear AurkA signal with one or two spots of Cep170 within the cells (<150 cells from four independent experiments). **(C)** Representative interphase with nuclear AurkA and intact nuclear envelope. **(D)** FACS panels show the DNA content of cells under indicated conditions (one [of three] representative experiment is shown); the percentage of cells in the G2 and M phases is indicated in the table below (mean ± SD, three independent experiments). Histograms represent the percentage of interphases with nuclear AurkA localization in control cultures and after the indicated treatments (at least 450 cells; three independent experiments). **(E)** Representative images, from a time-lapse experiment, of the dynamic accumulation of endogenous AurkA (Venus-tagged, hTERT RPE-1 cells) before mitotic entry (10 ≤ n ≤ 49 at different timepoints, because individual trajectories in this experiment were not synchronized; two experiments). The first observable frame in mitosis is set as time 0 (images and graph), and a time interval of 240 min before mitotic entry is analyzed for mean nuclear fluorescence (red plot; a.u., arbitrary units) and nucleo/cytoplasmic ratios (blue plot). In the shown example, nuclear accumulation is observable from t = −90 min. **(F)** Representative IF images for AurkA and PCNA staining in the control culture and upon indicated treatments. Histograms represent the percentage of cells displaying nuclear AurkA that are also positive for PCNA staining (arrowed in the control panel), in the indicated conditions (at least 150 cells; three independent experiments). Error bars: SD; ns: not significant; **$P < 0.001$; and ***$P < 0.0001$, chi-squared test. Scale bars: 10 $\mu m$ (A, B, C) or 20 $\mu m$ (E, F).

non-transformed cellular background and observed how this localization is influenced by the cell cycle phase and the nuclear export, whereas it is not affected by its catalytic activity. By AURKA overexpression studies, we then uncovered that AurkA increased levels alone are not sufficient to determine increased AurkA localization. Gene expression analyses revealed a strong co-overexpression of AURKA, TPX2, and exportin-2 CSE1L in tumors. Interestingly, when AURKA is overexpressed in non-transformed cells, its nuclear localization is increased upon TPX2 co-overexpression, or after proteasome inhibition, indicating involvement of AurkA protein stability. This TPX2 role is independent of its AurkA-activating function. Finally, we show that TPX2 contribution to AurkA nuclear localization has functional

consequences in MCF10A cells on tumor-related AurkA roles. Together, our results suggest that AURKA/TPX2 co-overexpression in cancer has an impact not only on mitotic but also on interphasic nuclear AurkA oncogenic functions.

# Results

## Cell cycle phase and protein export influence AurkA nuclear levels

To investigate how nuclear localization of AurkA is regulated, we first analyzed endogenous AurkA by immunofluorescence (IF) in

non-transformed hTERT RPE-1 cells (Fig 1). AurkA staining is evident in about one-third of interphases: a fraction of cells only displays a centrosomal signal, whereas AurkA is nuclear (either uniformly diffused between the nucleus and the cytoplasm or nuclear-enriched; see Fig 1A) in 20% of all interphases. Cells with nuclear AurkA display two spots of Cep170 (Fig 1B), characteristic of the G2 phase, and an intact nuclear envelope as assessed by lamin B1 staining (Fig 1C). Indeed, the percentage of cells with nuclear AurkA is consistent with the fraction of asynchronous hTERT RPE-1 cell population being in the G2 or M phases, and enriching the population in G2 by treatment with the Cdk1 inhibitor RO3306 induced a parallel increase in interphases with nuclear AurkA (Fig 1D). Video recording of hTERT RPE-1 cells expressing *AURKA* endogenously tagged with Venus (20) shows that an increase in AurkA nuclear levels begins about 90 min before mitotic entry, paralleled by a slight increase in the nucleo/cytoplasmic ratio (Fig 1E). Together, these observations indicate that in non-transformed cells AurkA nuclear localization occurs in G2.

Import/export processes regulating AurkA localization in interphase are not well clarified. Rannou et al (28) reported an increase in nuclear AurkA—exogenously expressed—after export inhibition. To investigate the relevance of active export of endogenous AurkA, we treated hTERT RPE-1 cells with leptomycin B (LMB), an inhibitor of the export receptor CRM1. Under these conditions, the cell cycle is not altered (FACS panels and table, Fig 1D) and AurkA is nuclear in about 40% of cells compared with 20% in control cultures (Fig 1D). These data suggest that active export may modulate the nuclear localization of the kinase. AurkA nuclear localization was not increased when we combined RO3306 with LMB treatments, compared to treatment with RO3306 alone (Fig 1D), suggesting that the export process is relevant for AurkA localization in a time window preceding G2. To confirm this hypothesis, cells were co-stained with the S-phase marker PCNA: results show that after LMB treatment, more than 40% of cells with nuclear AurkA are PCNA-positive (Fig 1F), compared with 20% in control cells. This increased association was not observed when cells were treated with RO3306, either alone or in combination with LMB, due to the enrichment of cells in the G2 phase. Indeed, the significant decrease in S-phase PCNA-positive cells after RO3306 treatment (Fig S1A) explains the lack of additive effects of the combined treatments (RO3306 + LMB) on AurkA nuclear localization (Fig 1D).

These results indicate that AurkA localization in the nucleus under physiological conditions strongly correlates with the G2 phase, and nuclear export contributes to limit AurkA nuclear accumulation in previous cell cycle phases.

## AURKA overexpression in non-transformed cells is not sufficient to yield its nuclear enrichment

To evaluate whether AURKA overexpression is sufficient to yield high levels of nuclear AurkA, which have been reported as oncogenic in cancer cells, we used an hTERT RPE-1 cell line that we have generated for stable and inducible expression of exogenous myc-tagged AurkA (Fig 2A (29)). In this cell line, after doxycycline (dox) induction, <60% of interphases are positive for AurkA staining (compared with 30% of hTERT RPE-1 cells staining positive for endogenous AurkA). Nonetheless, only about 15% of all cells

displayed AurkA nuclear localization (Fig 2B), a percentage comparable to control cultures (Fig 1D). Instead, a peculiar distribution rarely observed for endogenous AurkA in hTERT RPE-1 control cells was evident; that is, in about 50% of interphases, AurkA localized to the cytoplasm and appeared excluded from nuclei (Fig 2A and B). Inhibition of protein export by LMB determines a small, but significant, increase in cells with nuclear accumulation of AurkA (Fig 2B), indicating only a partial involvement of nuclear export in AurkA nuclear exclusion under overexpression conditions. Unexpectedly, the percentage of AurkA-positive cells decreased after LMB treatment (Fig 2B, left histograms); consistently, the overall amount of AurkA was reduced (Western blotting [WB] in Fig 2C). These observations suggest that inhibiting export from the nucleus, and entrapping AurkA therein, determines increased AurkA degradation, influencing the overall amount of the protein. To directly address this, we combined LMB with MG132 treatment to simultaneously inhibit nuclear export and proteasome degradation activity. Under these conditions, AurkA levels are higher (Fig 2C, compare the last two lanes) and cells with AurkA nuclear localization increase (90% of AurkA-positive cells) with an evident nuclear enrichment (Fig 2B). Interestingly, MG132 treatment alone was able to yield the same effect (Fig 2B), indicating that AurkA nuclear stabilization has a dominant effect over protein export. Comparable results were obtained by inhibiting the proteasome with epoxomicin (Fig S1B), suggesting that nuclear accumulation of AurkA is normally counteracted by protein degradation. FACS analyses indicated that neither MG132 nor epoxomicin treatment yielded major changes in cell cycle progression (Fig S1C). Given that AurkA is an APC/C-Fzr1 target (8, 30, 31) and Fzr1 is reported to be nuclear in interphase (32, 33), we investigated whether it is involved in AurkA nuclear degradation. Indeed, combined treatment with APC/C inhibitors Apcin and proTAME (34) in the AurkA-overexpressing cell line yielded a significant increase in cells with nuclear AurkA (45% compared with 15% in DMSO-treated cultures) and a corresponding decrease in AurkA nuclear-excluded cells (Fig 2D). To investigate whether this was a direct effect on the kinase, we assayed the localization of a non-degradable AurkA-deleted version, AurkAΔ67-Venus, lacking the A-box described as critical for APC/C-Fzr1 recognition (35). Nucleo/cytoplasmic ratio fluorescence measures after transient transfection in U2OS cells indicated that this deleted version is indeed nuclear-enriched compared with WT AurkA-Venus (Fig 2E).

In conclusion, we observed that overexpression of AURKA is not sufficient to increase nuclear localization of the kinase, which is negatively modulated by protein export and to a larger extent by protein degradation, suggesting that increased nuclear localization of AurkA in cancer cells depends on altered interphase regulatory mechanisms.

## AURKA, TPX2, and CSE1L are co-overexpressed in cancer

To investigate altered mechanisms that may be linked to the nuclear enrichment of AurkA in cancer, we first carried out an analysis on RNA-sequencing data coming from TCGA and GTEx cancer consortium projects, mining for genes encoding proteins involved in nucleus/cytoplasm shuttling with expression profiles in tumors similar to *AURKA*. To this aim, we selected from gene ontology (GO)

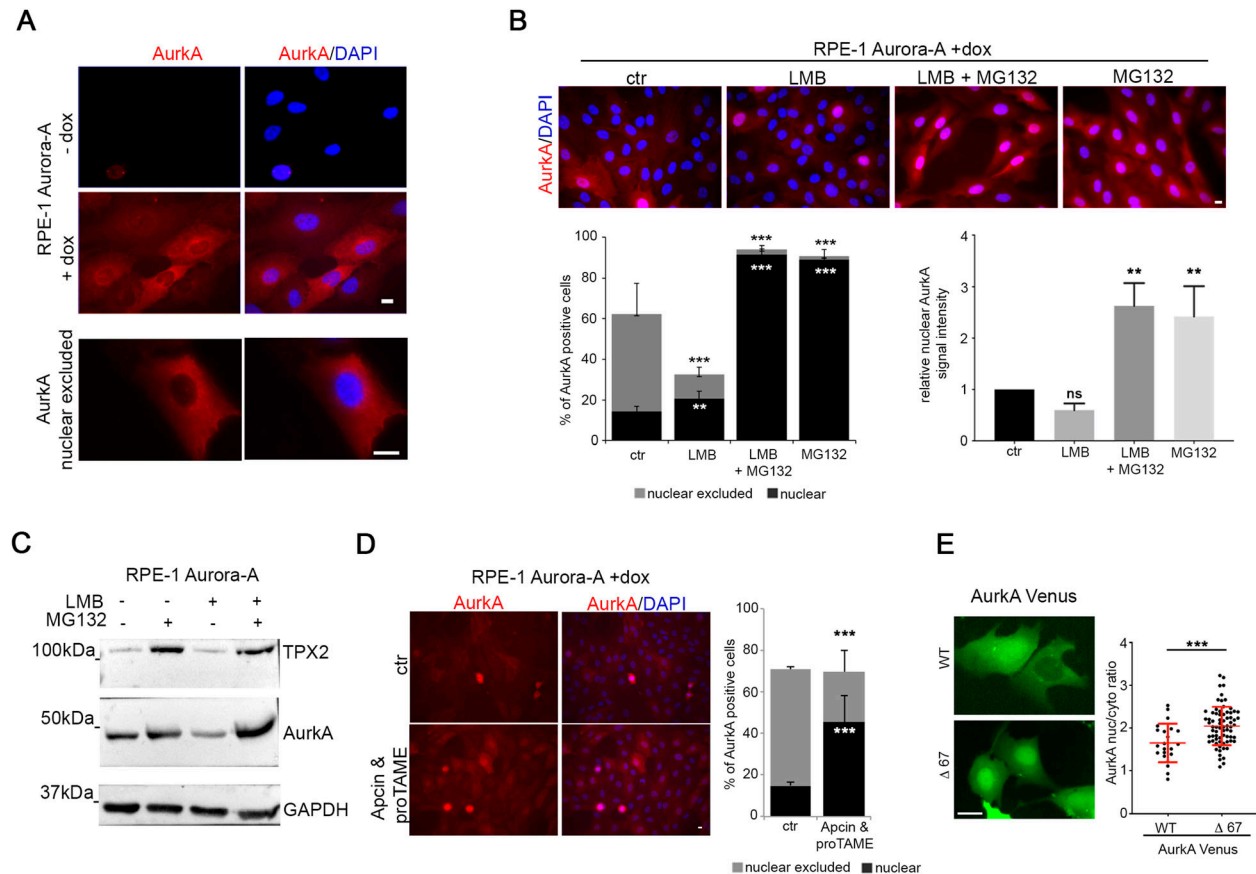

**Figure 2. Nuclear localization of overexpressed AurkA is counteracted by proteasome-dependent degradation.**
**(A)** IF panels show AurkA staining in the inducible Aurora-A–overexpressing hTERT RPE-1 cell line, ± doxycycline treatment. The lower panel shows a representative interphase in which AurkA is nuclear-excluded. **(B)** IF images show AurkA localization in Aurora-A–overexpressing hTERT RPE-1 cells (control or indicated treatments). Histograms on the left represent the percentage of nuclear or nuclear-excluded AurkA-positive cells (at least 600 cells per condition; three independent experiments), and those on the right, the mean intensity signal of nuclear AurkA (at least 300 cells per condition; three independent experiments; for each experiment, the control mean values have been considered as 1). **(C)** Immunoblotting for TPX2 and AurkA on lysates from Aurora-A–overexpressing cultures ± LMB and/or MG132 treatments; GAPDH is the loading control. **(D)** Representative IF images of AurkA staining in the Aurora-A–overexpressing hTERT RPE-1 cell line in control conditions (upper panels) or after treatment with APC/C inhibitors (lower panels): for the same conditions, the percentage of cells with nuclear or nuclear-excluded AurkA are indicated in the histograms (at least 600 cells per condition; three independent experiments). **(E)** Live-cell fluorescence images of AurkA-Venus WT and Δ67 localization in U2OS cells enriched in the G2 phase by 24 h of treatment with 1 μM RO3306. Graphs show the nucleus/cytoplasm ratio of AurkA WT and AurkA Δ67 signal intensity (each spot indicates a cell; at least 25 cells per condition; one [of three] representative experiment; mean values and SD are shown in red). Error bars: SD; ns: not significant; **P < 0.001; and ***P < 0.0001, chi-squared test (histograms in B [left] and D), and Kruskal–Wallis test or Mann–Whitney test (histograms in B [right] and dot plot in E). Scale bars: 10 μm.

all the genes associated with import/export from the nucleus and we computed, for each one, the pairwise expression correlation to *AURKA*, using Pearson's correlation analysis. Strikingly, we found a high correlation between exportin-2 (also known as chromosome segregation 1 like, *CSE1L*) and *AURKA* in multiple cancer types, with breast invasive carcinoma—where nuclear AurkA has been described (24, 25)—scoring the highest *P*-value (Fig 3A and Table S1). Exportin-2 mediates importin-alpha re-export from the nucleus to the cytoplasm, after NLS-containing import cargos have been released into the nucleoplasm, thus acting as a general regulator of nuclear import (36). The observed correlation suggests that AURKA overexpression in cancer associates with conditions of increased protein import. Interestingly, a similar correlation was observed for *CSE1L* and *TPX2* (Fig 3A and Table S1). Indeed, *AURKA* and *TPX2* are among the top five genes in the whole human genome most highly co-expressed in tumors together with *CSE1L*, with *AURKA* being the

top-ranking one (Fig 3B). Based on these results on the well-known nuclear localization of TPX2 in interphase (37, 38) and its frequent co-overexpression with AURKA in tumors (references 39, 40; see heatmap profiles and tissue-wise expression in different cancer types compared with normal tissues in Fig 3C and D), we hypothesized that TPX2 influences the AurkA nuclear pool and that it may directly contribute to the increased nuclear localization of the kinase observed in certain cancer types (22).

## TPX2 contributes to the accumulation of nuclear AurkA

To test this possibility, we first evaluated the correlation between AurkA and TPX2 levels in nuclei of non-transformed hTERT RPE-1 cells. A positive correlation of IF nuclear signals indicated that AurkA nuclear localization occurs in cells with high levels of TPX2 (Fig 4A). Consistently, we noticed that the nuclear enrichment of

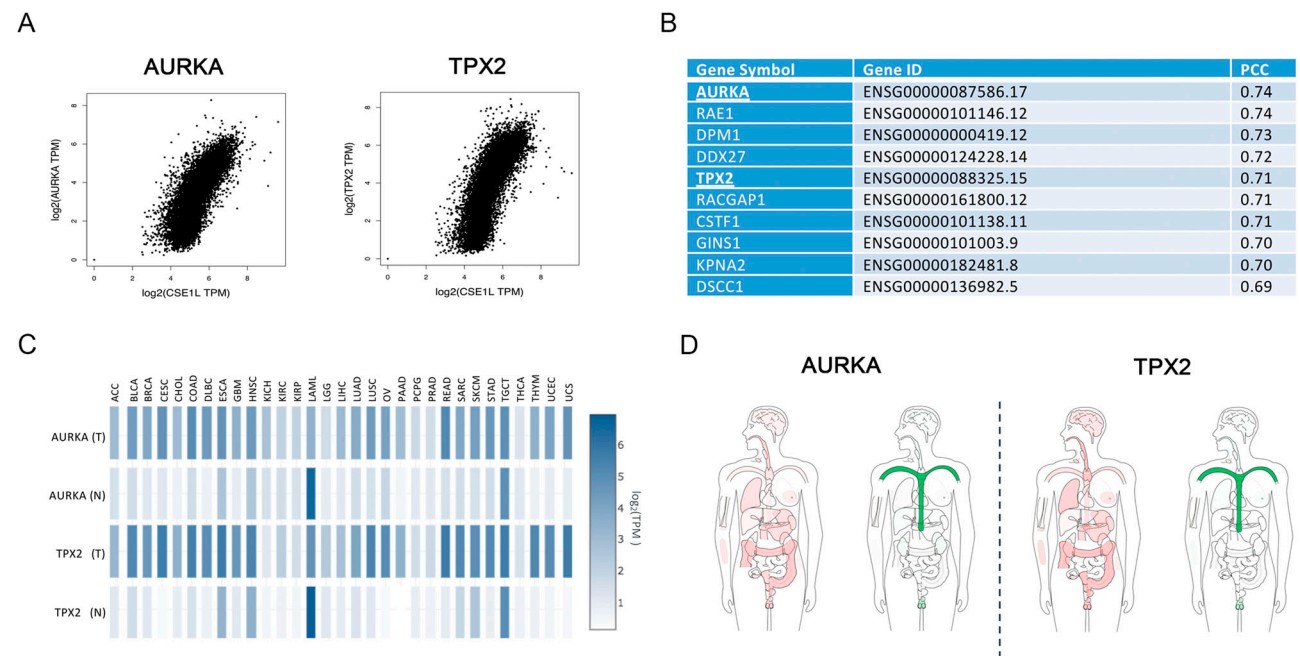

**Figure 3. Co-overexpression of AURKA, TPX2, and CSE1L in different tumor types.**
Expression analyses on RNA-sequencing data coming from TCGA and GTEx cancer consortium projects. **(A)** Pairwise gene expression correlation analysis of CSE1L with AURKA and TPX2 in tumors. The Spearman correlation values and corresponding *P*-values for each tumor type are reported in Table S1. **(B)** Table reports the list of the top 10 genes co-overexpressed with CSE1L. **(C)** Expression matrix plot (log₂ TPM) of AURKA and TPX2 in tumors (T) and normal tissues (N). **(D)** BodyMap of tissue-wise expression of AURKA and TPX2 in normal (green) and tumor (red) tissues.

AurkA observed in MG132-treated cells was accompanied by an accumulation of TPX2 (Fig 2C). Interestingly, we revealed AurkA/TPX2 in situ proximity ligation assay (*is*PLA) spots in nuclei of about 25% of interphases (Fig 4B) consistent with the percentage of cells displaying nuclear AurkA (Fig 1D) and indicating for the first time a nuclear interaction between the two proteins. To directly evaluate whether co-overexpression of TPX2 is able to increase the fraction of cells with nuclear-localized AurkA, we generated an additional hTERT RPE-1 cell line where both AURKA and TPX2 are overexpressed, in an inducible manner (Figs 4C and S2A). Both the percentage of cells displaying AurkA/TPX2 *is*PLA spots and the *is*PLA signal intensity significantly increase in the AURKA/TPX2-overexpressing cell line (Fig 4B). Importantly, after dox induction 60% of AURKA/TPX2-overexpressing interphases are AurkA-positive, and AurkA is enriched in cell nuclei in about 40% of interphases (Fig 4D), a fraction significantly higher compared with control and AURKA-overexpressing cell lines (see Figs 1 and 2). Treatment with LMB had no effect on either nuclear localization or AurkA levels (Figs 4D and S2B), supporting a key role of TPX2 in regulating the AurkA nuclear fraction. Again, when we treated these cells with MG132, both AurkA and TPX2 levels increased (Fig S2B) and the almost totality of cells (<90%) displayed nuclear AurkA (Fig 4D). As for the AURKA-overexpressing cell line, treatment with epoxomicin yielded comparable results (Fig S2C).

To further assess the importance of the AurkA/TPX2 interaction, we used mutant/truncated versions of the two proteins. First, we analyzed the localization of the Venus-tagged AurkA^S155R mutant, described to have an impaired interaction with TPX2 (41, 42) after transient transfection in U2OS cells, and observed a decreased nucleo/cytoplasmic ratio compared with AurkA WT signal (Figs 4E

and S2D). We then evaluated how AurkA localizes after co-overexpression in hTERT RPE-1 with a truncated version of TPX2 lacking the AurkA interaction region (aa 1–43 (11, 43)). To this aim, we generated an additional cell line for the stable and inducible co-overexpression of AURKA and TPX2Δ43 (Fig S2E). In this cell line, interphases with AurkA nuclear localization are comparable to control cultures and to the AURKA-overexpressing cell line (Fig 4F). Consistently, cells with AurkA cytoplasmic localization (nuclear-excluded) represent the majority of AurkA-positive interphases in the AURKA alone and AURKA/TPX2Δ43 cell lines, whereas they almost disappear (about 10% of the population) in the AURKA/TPX2 cell line (compare Figs 2B and 4D and F). Furthermore, only in the AURKA/TPX2-overexpressing cell line we observed an increase in PCNA-positive (i.e., S-phase) cells within the nuclear AurkA-positive population (Fig 4G).

Given the activating role of TPX2 on AurkA, we evaluated whether the catalytic activity of AurkA is required for its nuclear localization. We used 250 nM MLN8237 for 2 h to inhibit AurkA kinase activity (Fig 4H). The percentage of cells displaying AurkA nuclear localization in control hTERT RPE-1 or AURKA/TPX2-overexpressing cultures did not change upon MLN8237 treatment (compared with control cultures), both in asynchronous and in G2-enriched (RO3306-treated) cultures. To confirm that AurkA kinase activity is not required for its nuclear localization, we transfected in U2OS cells a kinase-deficient mutant (AurkA K162R) and showed that its nucleus/cytoplasm ratio is comparable to that of AurkA WT (Fig 4I).

Together, these results indicate an important function of TPX2 in regulation of AurkA interphase localization and suggest that it may play a role, independent of its activating function, in nuclear enrichment of the kinase in cancer.

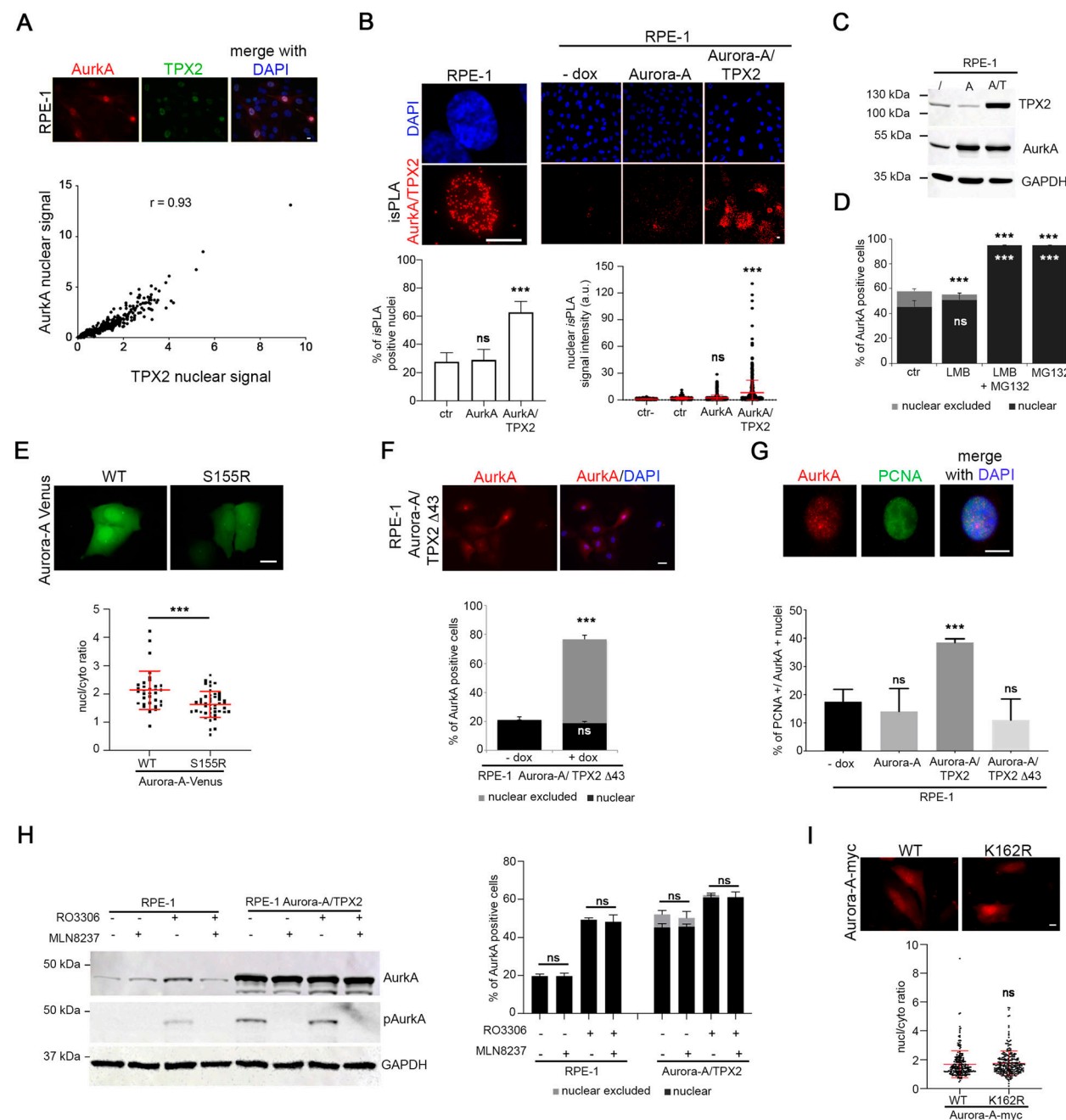

**Figure 4. TPX2 contributes to determine the AurkA nuclear fraction.**
**(A)** hTERT RPE-1 cells were stained for AurkA and TPX2 (IF panels), and correlation of respective signals was measured (<200 cells per condition; three independent experiments). **(B)** Representative example of AurkA/TPX2 *is*PLA signal in interphase hTERT RPE-1 cultures is shown on the left. Lower magnification representative images of AurkA/TPX2 *is*PLA signals in the indicated cell lines are shown on the right. Histograms represent the percentage of cells with nuclear AurkA/TPX2 *is*PLA spots, and the dot plot on the right represents the nuclear intensity of *is*PLA signals within the *is*PLA-positive population, in the indicated cell lines (a.u., arbitrary units; at least 460 total cells per condition were analyzed, from three independent experiments). Ctr- indicates a condition in which cells were incubated with only one primary antibody. **(C)** Western blot of TPX2 and AurkA in control (/) and in the inducible Aurora-A (A)– and Aurora-A/TPX2 (A/T)-overexpressing hTERT RPE-1 cell lines. GAPDH is a loading control. **(D)** Histograms represent the percentage of nuclear or nuclear-excluded AurkA-positive cells, for the indicated conditions (at least 300 cells per condition; three independent experiments). **(E)** Localization of transiently transfected AurkA WT and S155R mutant in U2OS cells enriched in the G2 phase by 1 $\mu$M RO3306 treatment for 24 h. Graphs show the signal intensity (nucleus/cytoplasm ratio) (at least 30 cells per condition; one [of three] representative experiment). **(F)** AurkA localization in the inducible Aurora-A/TPX2Δ43-overexpressing hTERT RPE-1 cell line (IF panels). Percentages of AurkA-positive cells with AurkA localization in the nucleus or nucleus-excluded are shown in the histograms (at least 600 cells per condition; three independent experiments). **(G)** IF of a PCNA-positive interphase displaying AurkA nuclear localization. Histograms indicate for each cell line the percentage of cells with PCNA staining displaying nuclear AurkA (<100 cells per condition; three independent experiments). **(H)** Western blot of TPX2, active p-Thr288-AurkA (pAurkA), and AurkA in control and Aurora-A/TPX2-overexpressing hTERT RPE-1 cells in the indicated conditions. GAPDH is a loading control. The corresponding percentage of nuclear or nuclear-excluded AurkA-positive cells is represented in the histograms on the right (700 cells per condition; three independent experiments). **(I)** Localization of transiently transfected AurkA WT and K162R mutant in U2OS cells. Graphs show the signal

## TPX2 promotes nuclear AurkA functions in a 3D breast cell–derived model

In order to evaluate whether TPX2 co-overexpression has functional consequences on processes described to be dependent on nuclear AurkA, we generated additional cell lines for the expression of AURKA, or AURKA/TPX2, in MCF10A breast non-transformed epithelial cells (Fig 5), based on the ongoing active investigation of the nuclear role of AurkA in breast cancer (see the Introduction section). We first confirmed the localization of AurkA, which is mostly nuclear-excluded in cells overexpressing AurkA alone (Fig 5A, central panels) and became mainly nuclear when AurkA is co-overexpressed with TPX2 (Fig 5A, right panels). It is well known that nuclear functions of AurkA correlate with an increase in transformed cells with breast cancer stem cell properties (24). Sphere-forming efficiency (SFE) and sphere size are used to assess the number of mammary cells with stem and EMT features, with SFE ranging from 0.1% to 0.7% in normal mammary epithelial cells, and from 1% to 3% in breast cancer cell lines (44, 45). Therefore, we moved to a mammosphere 3D condition, where we confirmed similar levels of AurkA in both AURKA- and AURKA/TPX2-overexpressing MCF10A cell lines by WB (Fig 5B) and RT–PCR (Fig 5C). We evaluated their sphere-forming ability by both counting the number and measuring the size of the obtained mammospheres. Both analyses demonstrate that AURKA overexpression increases mammosphere formation (Fig 5D); importantly, the effects are exacerbated when TPX2 is co-overexpressed. Accordingly, the stemness marker CD44, previously shown to increase upon nuclear AURKA overexpression (24), is up-regulated in AURKA-overexpressing conditions, and even more in AURKA/TPX2–co-overexpressing mammospheres (Fig 5E), supporting the relevance of TPX2 as a determinant of AurkA nuclear functions. As an independent measure of AurkA nuclear functions, we evaluated the expression of one of the reported up-regulated genes of the hypoxia signaling signature (27), that is, CXCR4, by RT–PCR in the mammospheres (Fig 5E). Consistent with our hypothesis, AURKA/TPX2 overexpression leads to a significant increase in mRNA expression with respect to AURKA overexpression alone. Overall, these data support the idea that overexpressed TPX2 plays a key role in mediating the oncogenic functions previously described for nuclear AurkA (24, 27).

# Discussion

In the present work, we studied the mechanisms that regulate AurkA localization in interphase nuclei, a condition that is associated with its oncogenic properties (22). An increased AurkA nuclear localization is reported in the literature as a prognostic marker for poor survival, particularly in breast cancer (23). Emerging evidence proposes that this AurkA fraction operates through both kinase-dependent and kinase-independent roles in cell transformation and cancer (24, 27), highlighting the importance to better understand its regulation.

Starting from physiological conditions, we report that nuclear AurkA localization becomes evident in G2 cells, about 90 min before mitotic entry, similar to what has been reported for exogenous AurkA (46). Interestingly, our data indicate a strong correlation between AurkA nuclear localization and high levels of its major regulator TPX2, which is nuclear in interphase (37). Most importantly, we revealed for the first time a nuclear interaction between AurkA and TPX2, in addition to the well-characterized interaction at spindle microtubules in mitosis (9), suggesting that TPX2 may be involved in the regulation of AurkA localization not only in mitosis, but also in G2 (Fig 5F). The interaction between TPX2 and AurkA in the nucleus may be a step required for correct mitotic entry. Interestingly, we recently reported that TPX2 and the spindle orientation regulator NuMA interact in interphase nuclei and at spindle poles (29), supporting the hypothesis that mitotic complexes start assembling in G2. An additional, not mutually exclusive, possibility is that the interaction between AurkA and TPX2 detected in interphase nuclei is linked to their emerging non-mitotic roles (19, 21, 38, 47).

Despite the increasing interest that nuclear localization of AurkA is raising, no canonical NLS or NES sequences within the kinase, nor post-translational modifications related to interphase localization, have been described. It has been reported that the nuclear localization determinants are within the C-terminus, whereas the N-terminus contains the cytoplasmic ones (24). Indeed, Rannou and colleagues proposed that overexpressed AurkA undergoes active export (28). Consistently, we observed that inhibiting CRM1 increases the fraction of cells with nuclear AurkA in a cell cycle window that precedes G2, suggesting that AurkA displays a shuttling behavior in the S phase that prevents premature activation of mitotic pathways.

Given the emerging importance of nuclear AurkA in tumorigenesis, we then asked whether increasing its expression directly correlates with this localization pattern. Surprisingly, we observed that under overexpression conditions in non-transformed cells AurkA is mostly cytoplasmic and excluded from nuclei. Thus, tumors where AURKA is overexpressed and nuclear may have concomitant alterations in pathways that control AurkA nucleus/cytoplasm trafficking. Our analyses of gene expression databases identify AURKA as the gene with the highest correlation with CSE1L in multiple cancer types, with breast invasive carcinoma data displaying the highest significance. CSE1L mediates the re-export from the nucleus to the cytoplasm of importin-alpha, the major import vector together with importin-beta for NLS-containing proteins, ensuring efficient recycling and nuclear import (36). Our results suggest therefore that in certain cancer types, AURKA overexpression correlates with increased import rates. Interestingly, the association data are similar for TPX2, which we propose as highly relevant for enriched AurkA nuclear localization (see below), and the three genomic loci (AURKA, TPX2, and CSE1L) are all in the 20q chromosome arm, frequently amplified in cancer.

Inhibiting nuclear export by LMB treatment modified the localization pattern of overexpressed AurkA; surprisingly, the treatment concomitantly reduced protein levels of the kinase. This latter effect was counteracted by proteasome inhibition, which was able

---

intensity (nucleus/cytoplasm ratio) (at least 230 cells per condition; three independent experiments). Error bars: SD; r: correlation value; ns, not significant; and ***P < 0.0001, chi-squared test (left histograms in (B); D-F-G-H), Kruskal–Wallis test ((B), dot plot on the right), and Mann–Whitney test (E, I). Scale bars: 10 μm.

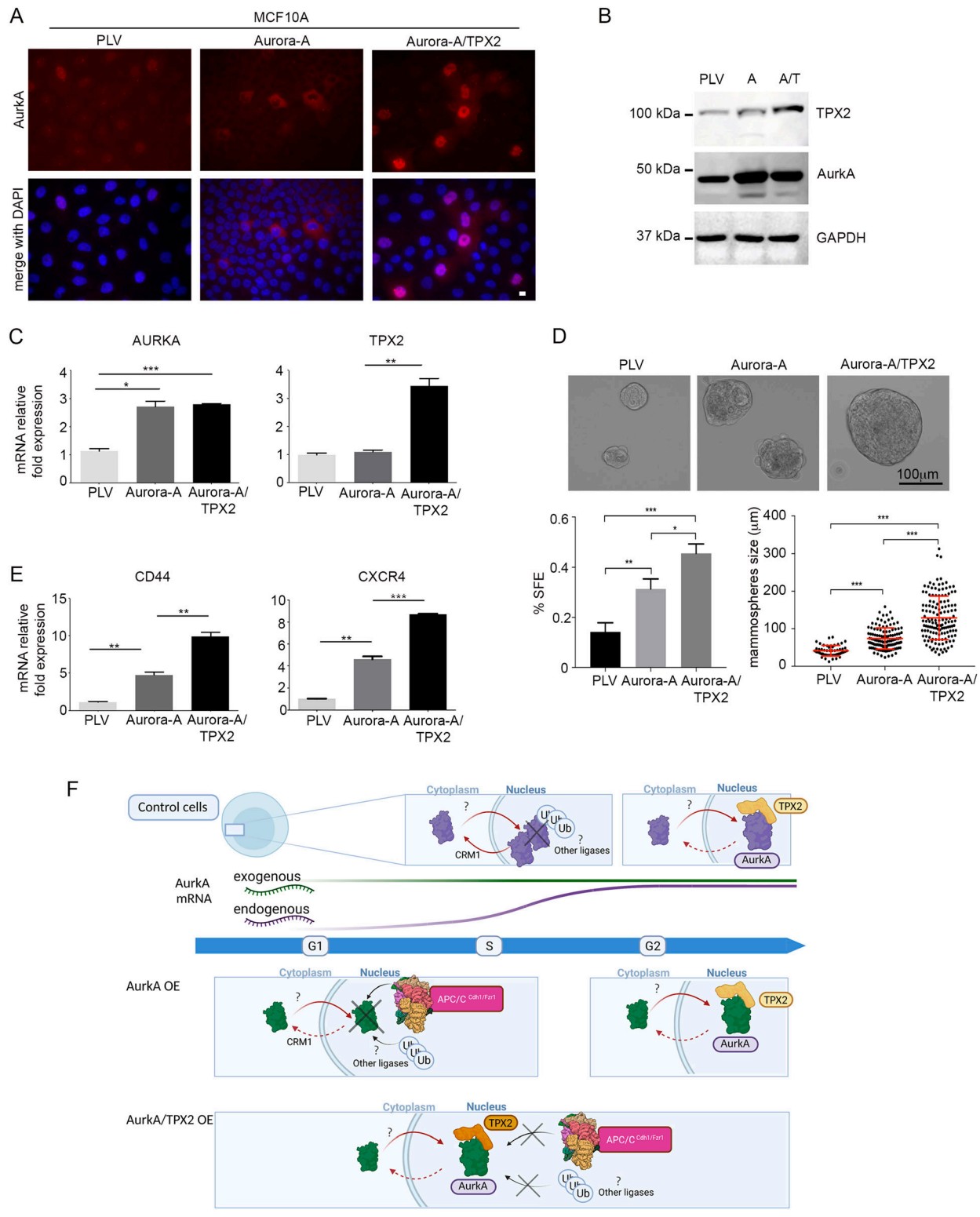

**Figure 5. TPX2 influences AurkA nuclear function in a 3D breast-derived model.**
**(A)** IF images of control (PLV), Aurora-A–, and Aurora-A/TPX2-overexpressing MCF10A cells. **(B)** Immunoblotting for TPX2 and AurkA on lysates from mammosphere cultures of control (PLV), Aurora-A (A)–, and Aurora-A/TPX2 (A/T)-overexpressing cell lines; GAPDH is the loading control. **(C)** Histograms show the average mRNA expression for the indicated genes (mean ± SEM; fold increases are from three independent experiments; TBP was used to normalize data). **(D)** Representative images of mammospheres formed from the three different cell lines. Histograms represent the sphere formation efficiency (SFE; at least 230 mammospheres per condition; three independent experiments). Dot plots indicate mammosphere size for the three cell lines (at least 50 mammospheres per condition; two independent experiments). **(E)** Histograms show the average mRNA expression for the indicated genes (mean ± SEM; fold increases are from three independent experiments; TBP was used to

per se to induce a strong accumulation of AurkA in virtually all interphase nuclei. These data suggest that premature entry of AurkA in the nucleus—when overexpressed—is prevented, in cell cycle phases when AURKA is normally not expressed, by a proteasome-dependent mechanism (Fig 5F). Of note, the ubiquitin ligase involved in AurkA degradation at mitotic exit—APC/C[Cdh1/Fzr1]—has been reported as nuclear and is active until the G1/S transition (32, 33). Consistently, using inhibitors of APC/C[Cdh1/Fzr1] we observed an increase in AURKA-overexpressing cells displaying nuclear localization. Thus, concomitant deregulation of ubiquitin ligases may be a route to increased levels of overexpressed AurkA in interphase nuclei. Evidence of lowered levels of Cdh1 in tumors, and specifically in breast cancer, exists (48, 49). Additional ubiquitin ligases may contribute to this interphase modulation. In this respect, it is interesting to note that ubiquitin ligases that modulate AurkA stability, for example, Fbxw7 and VHL, are frequently mutated in cancer (references 22, 50 and references therein). It will be interesting to further investigate the connection between AurkA protein stabilization and cancer, initially proposed when constitutive stabilizing phosphorylation of Ser51 was shown in head and neck cancer (51), in light of the contribution to its nuclear-localized oncogenic functions.

The AurkA nuclear enrichment observed upon proteasome inhibition may partially be the result of the stabilization of other factors, influencing in turn AurkA nuclear localization. TPX2 may be one such factor, because we observe that its levels increase after treatment with MG132. Given the frequent co-overexpression with AURKA in cancer (references 39, 52; updated data in Fig 3 of this study), TPX2 is therefore an interesting candidate to promote nuclear AurkA oncogenic functions. Indeed, our results indicate that TPX2 co-overexpression is able to increase the nuclear localization of AurkA, in a manner that depends on its ability to bind AurkA (Fig 5F). This effect does not require AurkA activity, thus not relying on the TPX2-activating function on AurkA. Further investigation will be required to clarify whether TPX2 is necessary for AurkA import into the nucleus, or whether it contributes to the nuclear accumulation of the kinase by binding it into the nucleus and protecting it from proteasome-dependent degradation and/or nuclear export. These results open the interesting possibility that co-overexpression of AURKA and TPX2 in cancer (39, 40) is associated not only with altered mitotic functions resulting in chromosomal instability, but also with emerging nuclear functions of AurkA. Among these, it has been shown that nuclear AurkA promotes mammosphere formation and transcriptional activation of myc- and FOXM1-regulated genes promoting a stemness phenotype, and of a hypoxia network signature (24, 25, 27). Supporting our hypothesis, MCF10A breast-derived cells engineered to overexpress AURKA and TPX2 yield an increase in the number and size of mammospheres, and

transcriptional activation of stemness (CD44) or hypoxia signaling (CXCR4) genes, compared with overexpression of AURKA alone. Still, the latter situation is already associated with a lower but significant increase compared with control conditions. This may be due to the fraction of cells displaying nuclear AurkA when overexpressed alone. In an alternative scenario, given the pleiotropic roles of AurkA (19) there may be additional routes, not associated with the nuclear fraction of AurkA, cooperating to generate a stemness phenotype.

Our observations that TPX2 contributes to nuclear oncogenic functions of AurkA, described at least in part as non-kinasic, also open interesting opportunities in the therapeutic field. Although targeting AurkA was initially attempted by developing ATP-competitive inhibitors, an approach that is still actively pursued (15), recent strategies include developing protein degraders (53, 54) and protein–protein interaction inhibitors (55, 56, 57, 58). Among the latter class, molecules that impair the formation of the AurkA/TPX2 complex may result in promising tools to disrupt AurkA nuclear functions in cancer, which would instead remain largely untargeted by molecules inhibiting its kinase activity.

In conclusion, understanding the regulation of nuclear AurkA may not only provide important information on its oncogenic functions but also drive the design of novel approaches to effectively target this pool of the kinase in specific tumor types where it has been shown to be relevant for cancer progression.

# Materials and Methods

### Cell cultures, synchronization protocols, and treatments

Cells were grown in complete DMEM (human U2OS cells) or DMEM/F-12 (human hTERT RPE-1 epithelial cell line, and hTERT RPE-1 cells modified to express endogenously tagged *AURKA* [RPE-1 AURKA-VenusKI] (20)) supplemented with 10% FBS. hTERT RPE-1 cell lines for the stable and inducible overexpression of myc-tagged AURKA, alone or in combination with either FLAG-TPX2 or FLAG-Δ43TPX2, were grown as the hTERT RPE-1 cell line, using tetracycline-free FBS. 1 $\mu$g/ml dox for 24 h was used for induction.

Human breast MCF10A cells were grown in HuMEC Basal Serum-Free Medium supplemented with the HuMEC Supplement Kit, as described in reference 59.

When indicated, cells were treated as follows: (a) 6 $\mu$M RO3306 (SML0569; Sigma-Aldrich) for 18 h; (b) 10 $\mu$M MG132 (SC-201270; Santa Cruz Biotechnology) or 10 $\mu$M epoxomicin (BU-4061T; Selleck Chemicals) for 4 h; (c) 20 nM LMB (ALX-380-100-C100; Enzo Life Sciences) for 2 h (IF) or 4 h (WB); (d) 20 $\mu$M Apcin (#5747; Bio-Techne) and 40 $\mu$M proTAME (I-440-01M; R<D Systems) for 6 h; and (e) 250 nM MLN8237 (alisertib; Selleck Chemicals) for 2 h.

normalize data). Error bars: SD; *$P < 0.01$; **$P < 0.001$; and ***$P < 0.0001$, *t* test (C, D, E) or Kruskal–Wallis test (D). Scale bars: 10 $\mu$m (A) and 100 $\mu$m (D). **(F)** Schematic model of the mechanisms that regulate AurkA localization in interphase nuclei. In control cells (top), AurkA nuclear accumulation in the S phase is prevented by the nuclear export protein CRM1; proteasome-dependent degradation may also contribute. In G2 cells, the interaction between AurkA and TPX2 promotes AurkA nuclear localization. Under AurkA overexpression conditions (AurkA OE, middle panels), AurkA nuclear accumulation in the G1 and S phases is limited by the action of APC/C[Cdh1/Fzr1] and other ubiquitin ligases. These effects are counteracted in cells that overexpress both AurkA and TPX2 (AurkA/TPX2 OE, lower panels), by their interaction, thus promoting the accumulation of nuclear AurkA. The blue arrow represents cell cycle progression, and the two colored lines represent AurkA mRNA levels (exogenous: green; endogenous: purple). Created with BioRender.

## Mammosphere cultures

Single-cell suspensions of MCF10A (and derived; see below) cell lines were grown in ultralow attachment six-well plates (Corning) at a density of 4,000 cells/ml as described in reference 60. After 10 d, the diameters of mammospheres were measured in phase-contrast pictures (ZOE) using ImageJ software. Numbers of mammospheres (diameter >50 $\mu$m) were counted, and the efficiency of mammosphere formation was evaluated (%SFE = number of mammospheres/number of plated cells * 100). The mammosphere pellet was collected by gentle centrifugation (300$g$, 5 min) for RNA and protein extraction.

## Real-time PCR

Total RNA was extracted and reverse-transcribed as described in reference 60. Primers were designed as follows:

AURKA FW 5′-TTGAACACCCCTGGATCACA-3′
AURKA RV 5′-GTCCAGCTCGACCAGGATG-3′
TPX2 FW 5′-AGATCGCCTGGAGAATTCGA-3′
TPX2 RV 5′-GGGGCATCATAGGAATAAGAGC-3′
hCXCR4 FW 5′-AATAAAATCTTCCTGCCCACC-3′
hCXCR4 RV 5′-CTGTACTTGTCCGTCATGCTTC-3′
CD44 FW 5′-CCAGAAGGAACAGTGGTTTGGC-3′
CD44 RV 5′-ACTGTCCTCTGGGCTTGGTGTT-3′
TBP FW 5′-TGCCCGAAACGCCGAATATAATC-3′
TBP RV 5′-TGGTTCGTGGCTCTCTTATCCTC-3′

Relative quantification was performed by the comparative cycle threshold method (61). The mRNA expression values were normalized to those of the TBP (TATA-box–binding protein) gene used as an endogenous control. One control of MCF10-derived mammospheres infected with the control vector (PLV) was randomly chosen as a control calibrator.

## Generation of stable cell lines

Cell lines for the stable and inducible overexpression of myc-tagged Aurora-A in combination with either FLAG-TPX2 or FLAG-Δ43TPX2 were generated using the previously described procedure (62) and plasmids listed below.

Aurora-A and TPX2 were overexpressed in MCF10A cells by lentivirus-mediated expression using lentivirus produced in HEK293T cells by co-transfecting lentiviral vectors with sequences for Aurora-A and TPX2 [VectorBuilder, vectors IDs: VB900000-1420ztx (pLV [Exp]-Puro-CMV<hAURKA) and VB900120-6505ssf (pLV[Exp]-Bsd-EF1A<hTPX2)], together with respective plasmids encoding for gag-pol and VSV-G proteins. The viral supernatant was collected 48 h post-transfection, filtered through a 0.45 $\mu$m pore size filter, and added to the cells in the presence of 2 $\mu$g/ml polybrene.

## Plasmids and transient transfections

Plasmids (epB_BSD_TT_Aurora-A-myc; epB_Puro_TT_FLAG-TPX2; epB_Puro_TT_FLAG-Δ43TPX2; and the hyPB7 plasmid encoding the hyperactive PiggyBac transposase) used for the generation of the stable transgenic hTERT RPE-1 cell lines were previously described (29, 62, 63). pVenus-N1-AURKA WT and mutant versions are previously published plasmids (54), with all cloning details available on request. Transient transfections of pVenus-N1-AURKA plasmids were carried out using a Neon electroporator (Thermo Fisher Scientific) according to the manufacturer's recommendations, and transfected cells were seeded onto eight-well microscope slides (Ibidi) for live-cell fluorescence microscopy.

pBK-CMV-myc-Aurora-A and pBK-CMV-myc-Aurora-A-K162R (kind gift of EA Nigg (64)) plasmids were transiently transfected in U2OS cells using Lipofectamine 2000 (Invitrogen Corporation, Thermo Fisher Scientific) following the manufacturer's instructions. The pBK-CMV empty vector was transfected in control cultures. Cells were harvested 30 h after transfection and analyzed by IF.

## Immunofluorescence

Cells grown on coverslips were fixed using 3.7% formaldehyde/30 mM sucrose in PBS for 10 min and permeabilized in PBS containing 0.1% Triton X-100 for 5 min; blocking and incubations with antibodies were performed in PBS with 0.05% Tween-20 and 3% BSA at room temperature. For PCNA staining, blocking and incubations were performed in PBS with 0.3% Triton X-100 and 3% BSA. Cells were counterstained with 4′,6-diamidino-2-phenylindole (DAPI, 0.1 $\mu$g/ml; Sigma-Aldrich) and mounted using Vectashield (Vector Laboratories). Primary antibodies were as follows: mouse anti-Aurora-A (610939, 0.5 $\mu$g/ml; BD Transduction Laboratories); rabbit anti-TPX2 (NB500-179, 1:1,500; Novus Biologicals); rabbit anti-Cep170 (final bleed, 1:100 (65)); rabbit anti-lamin B1 (ab16048, 1 $\mu$g/ml; Abcam); rabbit anti-PCNA (ab18197, 2 $\mu$g/ml; Abcam); and mouse anti-myc tag (clone 4A6, 05-724, 2 $\mu$g/ml; Merck Millipore).

Fixed samples were analyzed using (i) Nikon Eclipse 90i microscope equipped with the Qicam Fast 1394 CCD camera (QImaging) and 20X (N.A. 0.5), 40X (N.A. 0.75), or 100X (oil immersion; N.A. 1.3) objectives; (ii) Nikon Eclipse Ti inverted microscope, using a 60X (oil immersion, N.A. 1.4) objective and the Clara camera (ANDOR Technology); and (iii) Nikon Eclipse Ti2 inverted microscope equipped with the Kinetix sCMOS camera (Photometrix) and a 60X (oil immersion, N.A. 1.4) objective. Acquisitions were performed along the z-axis as follows: total range between 4 and 6 $\mu$m, 0.6 $\mu$m z-step (100X objective); total range between 3 and 6 $\mu$m, 0.3 or 0.9 $\mu$m z-step (60X objectives [iii] and [ii], respectively); and total range of 3 $\mu$m, 1.5 $\mu$m z-step (20X objective). Acquisitions, elaboration, and processing were performed using NIS-Elements AR or H.C. (Nikon) with the JOBS module for automated acquisitions, and Adobe Photoshop CS 8.

## *is*PLA

*is*PLAs were performed on cells grown on coverslips and fixed with formaldehyde (see the Immunofluorescence section) using the Duolink PLA kit (DUO92007 or DUO92008; Sigma-Aldrich) according to the manufacturer's instructions. The amplification time has been set to 60 min. The primary antibody pair to detect the interaction was mouse anti-Aurora-A/rabbit anti-TPX2, and DNA was stained with DAPI (see the Immunofluorescence section for details and acquisition procedures).

## Time-lapse video recording

Time-lapse imaging was carried out using cells seeded onto Ibidi eight-well slides and imaged at 37°C in L-15 medium/10% FBS using a 40X N.A. 1.3 oil immersion objective. Images were acquired on an automated epifluorescence imaging platform consisting of an Olympus IX81 inverted microscope (Olympus Life Science) equipped with a LED illumination source and a motorized stage and fitted with an incubation chamber. Image acquisition was controlled by Micro-Manager (66) and images were exported as tiff files.

## Image analysis and quantification

Images in Figs 2E and 4E were analyzed in ImageJ, with net green intensity (after background subtraction) in ROIs of 25 pixel radius from the nucleus and the cytoplasm used to calculate the nucleus/cytoplasm ratio of fluorescence.

Images in Figs 2B and 4A and B were quantified as follows: the "general analysis" module of NIS-Elements H.C. 5.11 was used for automatic recognition of the nuclei defined by the DAPI signal in all images. Nuclear signal intensity for AurkA, TPX2, or AurkA/TPX2 *is*PLA was then measured on maximum intensity projections from acquired z-stacks, after correction for external background. The AurkA-myc nucleus/cytoplasm ratio of fluorescence was calculated from the average values of signals within the nucleus and a cytoplasmic ROI. A threshold to automatically identify positive cells (*is*PLA or AURKA WT/K162R-transfected) was set at two standard deviations above the mean of the negative control conditions (*is*PLA negative control or PBK empty vector–transfected cultures).

## WB

Cell lysis and WB were performed as described in reference 62. Antibodies were as follows: rabbit anti-TPX2 (1:1,000; Novus Biologicals), mouse anti-Aurora-A (0.5 $\mu$g/ml; BD Transduction Laboratories), rabbit anti-pAurora-A (Thr288) (3079 clone C39D8, 1:100; Cell Signaling Technology, Inc.), and mouse anti-GAPDH (SC-32233, 1:1,000; Santa Cruz Biotechnology). HRP-conjugated secondary antibodies (Bio-Rad Laboratories S.r.l.) were revealed using the Clarity Western ECL Substrate (Bio-Rad Laboratories S.r.l.).

## FACS analysis

For FACS analysis, cells were collected by centrifugation, washed with PBS, and fixed with 50% methanol overnight at 4°C. Cell cycle phase distribution was analyzed after incubation for 30 min in the dark with propidium iodide (PI, P4170, 0.03 mg/ml; Sigma-Aldrich) and RNase A (0.2 mg/ml) using a flow cytofluorometric (BD FACS-Calibur) apparatus. Cell debrides and aggregates were gated out on the biparametric graph FL2-A/FL2-W. At least 15,000 events per sample were acquired. The percentage of cells in the different phases of the cell cycle was calculated using Floreada.io software.

## Expression analysis

The mRNA expression of *AURKA*, *TPX2*, and their co-expressed genes in cancers was obtained using Gene Expression Profiling Interactive Analysis API, to request the expression analysis from 9,736 tumors and 8,587 normal samples from TCGA and the GTEx data projects (67). Proteins involved in nucleus/cytoplasm shuttling, with expression patterns similar to AURKA and TPX2 across different tumors, were selected from GO (GO: 0006611, 65 genes; GO: 0006606, 177 genes). Pairwise expression correlation was assessed using Pearson's correlation analysis.

## Statistical analyses

Data were statistically analyzed using the InStat3 GraphPad 7 by (i) unpaired $t$ tests and ordinary one-way ANOVA multiple comparison tests for measurements of continuous variables; when samples were not normally distributed, the Mann–Whitney and Kruskal–Wallis tests, respectively, were used instead; (ii) chi-squared (and Fisher's exact) tests, in the contingency table analyses for measurements of categorical variables; and (iii) Spearman's (non-normally distributed samples) correlation for single-cell correlation analysis. The number of replicates and sample size are indicated in the corresponding figure legends. The criterion for statistical significance (*) was set at $P < 0.01$.

# Supplementary Information

# Acknowledgements

We thank Andrew Ying who wrote the ImageJ macro for image analyses, Enrico Cundari and Daniela Trisciuoglio for support with FACS analyses, Pietro Cirigliano for technical help with imaging experiments at the IBPM-Nikon reference center, and Rosario Luigi Sessa for experimental support. Research was supported by AIRC (IG-2021 ID: 25648 to G Guarguaglini and MFAG-2017 ID: 20447 to A Paiardini); Regione Lazio (PROGETTI DI GRUPPI DI RICERCA 2020 project ID: A0375-2020-36597 to G Guarguaglini); the Royal Society (International Exchange grant IES/R3/170195 to C Lindon); and Biotechnology and Biological Sciences Research Council (BBSRC) award (BB/R004137/1 to C Lindon). The studentship to C Ascanelli was funded by AstraZeneca UK. The implementation of the microscopy infrastructure was supported by MUR (Ministry of University and Research) funds to the PON project "IMPARA, Imaging from molecules to the preclinics".

## Author Contributions

IA Asteriti: conceptualization, supervision, investigation, visualization, and writing—original draft, review, and editing.
F Polverino: conceptualization, resources, formal analysis, supervision, investigation, visualization, and writing—review and editing.
V Stagni: resources, investigation, and writing—review and editing.
V Sterbini: investigation.
C Ascanelli: investigation.
FD Naso: resources.
A Mastrangelo: investigation.
A Rosa: resources.
A Paiardini: formal analysis, supervision, funding acquisition, investigation, visualization, and writing—original draft, review, and editing.

C Lindon: conceptualization, supervision, funding acquisition, and writing—review and editing.

G Guarguaglini: conceptualization, supervision, funding acquisition, project administration, and writing—original draft, review, and editing.

## Conflict of Interest Statement

The authors declare that they have no conflict of interest.

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
