## [Reviewer comments · Life Science Alliance]

AurkA nuclear localization is promoted by TPX2 and counteracted by protein degradation

Italia Asteriti, Federica Polverino, Venturina Stagni, Valentina Sterbini, Camilla Ascanelli, Francesco Naso, Anna Mastrangelo, Alessandro Rosa, Alessandro Paiardini, Catherine Lindon, and Giulia Guarguaglini

DOI: <https://doi.org/10.26508/lsa.202201726>

Corresponding author(s): *Giulia Guarguaglini, National Research Council and Italia Asteriti, National Research Council*

Review Timeline:	Submission Date:	2022-09-16
	Editorial Decision:	2022-10-10
	Revision Received:	2023-01-25
	Editorial Decision:	2023-01-27
	Revision Received:	2023-02-01
	Accepted:	2023-02-01

Transaction Report:

October 10, 2022

Re: Life Science Alliance manuscript #LSA-2022-01726-T

Dr. Giulia Guarguaglini
IBPM, CNR
Laboratories of genetics
Via degli Apuli 4
Via degli Apuli, 4
Rome 185
Italy

Dear Dr. Guarguaglini,

Thank you for submitting your manuscript entitled "AurkA nuclear localization is promoted by TPX2 and counteracted by protein degradation" to Life Science Alliance. The manuscript was assessed by expert reviewers, whose comments are appended to this letter. We invite you to submit a revised manuscript addressing the Reviewer comments.

Thank you for this interesting contribution to Life Science Alliance. We are looking forward to receiving your revised manuscript.

Sincerely,

Eric Sawey, PhD
Executive Editor
Life Science Alliance
<http://www.lsa-journal.org>

B. MANUSCRIPT ORGANIZATION AND FORMATTING:

Reviewer #1 (Comments to the Authors (Required)):

The paper from Asteriti and colleagues is from a leading team studying the AURKA/TPX2 'holoenzyme'. Using a variety of cell biology approaches, AurA nuclear localization was analysed as a function of cell cycle, and effects of overexpression (without stoichiometric TPX2) with relevance to interphase (when the nuclear envelope is intact). Co-expression of AURKA and TPX2 generates enhanced interphase nuclear localisation, and nuclear export appears to control levels to a certain extent; a role for proteasome-mediated degradation is also inferred, based on proteasome inhibition with MG-132. Informatics analysis shows co-overexpression of AurA and TPX2 and Exportin 2 (CSE1L) in some cancers, and the concept of the AurA/TPX2 complex being oncogenic is extended into an MCF10 mammosphere model.

The experiments performed support the conclusions of the manuscript, and the data are convincing and well presented overall, with good use of a S155R AURKA mutant and appropriate statistical analysis. Potential kinase-independent roles of AURKA are mentioned in the introduction, but are not followed up in the dataset, although they are mentioned in the context of non-mitotic roles (perhaps in the context of NMYC, see below). This is potentially important.

Major point 1: Does the nuclear accumulation established with AURKA alone or a AURKA/TPX2 complex require AURKA catalytic activity (ie, what happens with a kinase-dead AURKA)? This might distinguish it from the mitotic vs interphase biology that is emerging (as cited in the manuscript)

Minor point 1: Top of page 8. The sentence appears incomplete 'Based on these results, the well-known nuclear localization of TPX2 in interphase [36,37] and its frequent co-overexpression with AURKA in tumors ([38,39], add, see heatmap profiles and tissue-wise expression in different cancer types compared to normal tissues in Fig. 3C-D)'

Reviewer #2 (Comments to the Authors (Required)):

The manuscript from Asteriti and colleagues deepens our understanding of the nuclear accumulation of the cell cycle kinase AURKA. While AURKA was known to accumulate in the nucleus, the molecular mechanisms underlying its recruitment and the physio-pathological consequences remain largely unknown. By addressing these questions, this is an important manuscript which will certainly have an impact in the field.

The manuscript is very well written, concise yet precise. The data presented are convincing and are properly quantified. The amount of information provided in the materials and methods ensures the reproducibility of the findings by independent groups. There are few issues, mostly minor, which should be addressed prior to publication.

- Fig. 1B: Cep170 is supposed to localize at centrosomes only. Could the authors explain why we see an evident, network-like structure in the Cep170 micrographs?

- Fig. 1D: Why are there two control situations represented in the FACS profiles, and why is the content of G2/M in controls changing from 18% to 30%? This is far from being a negligible difference, and it is surprising that these cells have 18% of cells in G2/M in basal conditions already. Can the authors comment on these two aspects? In addition, a switch from 18% to 26% upon addition of LMB is far from being a negligible difference, so I would suggest toning down the statement "Under these conditions (LMB treatment), the cell cycle is not altered."

- Fig. 1E: the nuclear signal is evident even before the first 30 min. Potentially due to a cell cycle-related regulation of AURKA abundance, I agree that its levels seem to be overall lower at t=0 than at later timepoints. Nevertheless, it is hard to conclude on enrichment without providing a proper quantification such as nuclear/cytoplasmic ratio over time. I believe that this quantification should be provided.

- Fig. 2B: the authors describe that "the percentage of AurKa positive cells decreased after LMB treatment". This does not correspond to the micrograph shown, where we see a global increase of AURKA-positive signal in cells compared to the control condition and regardless of the localization of the kinase. How do authors explain that? This is somehow puzzling when looking at the blot in fig. 2C, where we see a loss of AURKA upon LMB treatment. Could it be that a specific fraction of AURKA - that remains visible on the image - is lost in western blotting analyses?

- MG132 is a non-specific inhibitor of the proteasome, and under some conditions it is known to trigger proteasome-independent effects such as autophagy induction. Can the authors corroborate their analyses with a more specific inhibitor such as exopomycin? Plus, could the MG132 effect on the number of cells with nuclear AURKA be induced by its indirect effect on the cell cycle rather than a specific effect on AURKA? What is the cell cycle profile of MG132-treated cells in the experiments performed in this manuscript?
- The authors state the fact that protein degradation is an important negative regulator of overexpressed AURKA, but they do not exclude that it is the same when the kinase is expressed at endogenous levels. Basically, are the mechanisms the same in the two conditions or not? In a cancer-related perspective, I believe it would be important to differentiate between these two conditions by assessing the behavior of the kinase with same parameters overall.
- Fig. 4B: please provide quantifications of PLA spots.
- Can the authors explain why cells co-expressing AURKA and TPX2 show no differences in the % of cells positive for AURKA upon LMB treatment? Could this be due to alterations in the cell cycle profile compared to AURKA-only overexpressing cells? Is this due to the direct interaction of AURKA and TPX2 in the nucleus which becomes stronger upon co-expression, thereby retaining AURKA in the nucleus? Similarly, can AURKA S155R still interact with TPX2, and conversely, can TPX2 43 still interact with AURKA?
- Fig. 5B: There is an increased abundance of TPX2 in AURKA-only overexpressing cells. Can the authors comment on that? It could be that there is a feedback loop to increase TPX2 stability under these conditions, but that since there is no difference at the mRNA level, TPX2 becomes the limiting factor to achieve a full expansion of mammospheres (such as the one seen upon co-expression).
- I somehow disagree with the following statement "these data support the idea that overexpressed TPX2, by increasing the levels of nuclear overexpressed Aurka, plays a key role in mediating the oncogenic functions described for nuclear Aurka". The authors do not rule out the possibility that TPX2 is not needed to increase AURKA levels, but to fully activate it instead. This is also visible on the blot in Fig. 5B, where the co-expression seems to slightly lower AURKA levels instead of increasing its abundance. Either they demonstrate that upon TPX2 co-expression, overexpressed nuclear AURKA needs TPX2 to get fully active, either I would suggest rephrasing this conclusion.
- I just wanted to point out that fluorescent micrographs are sometimes of bad quality (e.g. Fig. 1A, 2A, 2B). However, this could be due to the PDF conversion.

We thank both reviewers for their positive comments and for the suggestions that have helped us to improve the clarity and completeness of our manuscript. Our point-by-point reply follows.

Reviewer #1

Major point 1: Does the nuclear accumulation established with AURKA alone or a AURKA/TPX2 complex require AURKA catalytic activity (ie, what happens with a kinase-dead AURKA)? This might distinguish it from the mitotic vs interphase biology that is emerging (as cited in the manuscript)

We thank the reviewer for raising this issue thus enabling us to add an important piece of information to our manuscript. We addressed it by two distinct approaches:

1. We used the AurkA inhibitor MLN8237, both in non-transfected hTERT-RPE1 cells and in the AurkA/TPX2 overexpressing cell line, and evaluated the fraction of interphases displaying nuclear AurkA, in asynchronous or G2-enriched cultures. Results are shown in Figure 4H and indicate that there is no difference between control and MLN8237-treated cultures.
2. We transfected in U2OS cells expression constructs coding for myc-tagged AurkA wt or K162R (kinase dead). Analyses are included in Fig. 4I and show that the nucleo/cytoplasmic ratio (myc-tag signal) is comparable under the 2 conditions.

Minor point 1: Top of page 8. The sentence appears incomplete 'Based on these results, the well-known nuclear localization of TPX2 in interphase [36,37] and its frequent co-overexpression with AURKA in tumors ([38,39], add, see heatmap profiles and tissue-wise expression in different cancer types compared to normal tissues in Fig. 3C-D)'

We agree that the sentence construction we had used was misleading and we have modified it. We hope that the sentence now reads clearer.

Reviewer #2 (Comments to the Authors (Required)):

The manuscript from Asteriti and colleagues deepens our understanding of the nuclear accumulation of the cell cycle kinase AURKA. While AURKA was known to accumulate in the nucleus, the molecular mechanisms underlying its recruitment and the physiological consequences remain largely unknown. By addressing these questions, this is an important manuscript which will certainly have an impact in the field. The manuscript is very well written, concise yet precise. The data presented are

convincing and are properly quantified. The amount of information provided in the materials and methods ensures the reproducibility of the findings by independent groups.

There are few issues, mostly minor, which should be addressed prior to publication.

- Fig. 1B: Cep170 is supposed to localize at centrosomes only. Could the authors explain why we see an evident, network-like structure in the Cep170 micrographs?

In this study we used the anti-Cep170 unpurified serum, as indicated in the Methods section, which may account for some difference from our previously reported images using the purified antibody (Guarguaglini et al., 2005). Cep170 is reported as able to bind MTs in interphase too (Welburn et al., 2012 and our unpublished data) and this fraction may vary in different cell lines, so we believe that the signal observed with the unpurified serum in the hTERT RPE1 cell line may reflect MT-associated Cep170, but it is out of the scope of this study to analyze the extent and significance of this localization pattern. For the purpose of this manuscript, the “one spot” versus “two spots” staining (discriminating the G2 cells) is the relevant information and we agree that the shown micrographs may have looked misleading. Therefore, we have modified Figure 1B as follows: we have (i) replaced the “one spot” micrograph with an example which we hope better represents the relevant signal and (ii) used a distinct algorithm for deconvolution to improve the quality of both images.

- Fig. 1D: Why are there two control situations represented in the FACS profiles, and why is the content of G2/M in controls changing from 18% to 30%? This is far from being a negligible difference, and it is surprising that these cells have 18% of cells in G2/M in basal conditions already. Can the authors comment on these two aspects? In addition, a switch from 18% to 26% upon addition of LMB is far from being a negligible difference, so I would suggest toning down the statement "Under these conditions (LMB treatment), the cell cycle is not altered."

We agree with the reviewer that the two control situations shown in the FACS panels in Fig.1D were not comparable; therefore, the LMB sample, which had no significant difference from its own control (lower row in original Figure), appeared indeed different from the RO3306 control (upper row in the original Figure). Prompted by this and subsequent comments from the reviewer, we have now repeated a FACS analysis of cultures under all treatments and have included in main Figure 1D and Supplementary Figure 1C representative FACS panels and quantifications from triplicate experiments. New FACS data in Figure 1D (ctr, RO3306, LMB and the newly added panel RO3306+LMB) confirm that the control cultures display about 20% of G2/M cells (by PI incorporation) and that no significant difference is induced by LMB treatment, while RO3306 treatment increases the fraction of G2/M cells.

- Fig. 1E: the nuclear signal is evident even before the first 30 min. Potentially due to a cell cycle-related regulation of AURKA abundance, I agree that its levels seem to be overall lower at t=0 than at later timepoints. Nevertheless, it is hard to conclude on enrichment without providing a proper quantification such as nuclear/cytoplasmic ratio over time. I believe that this quantification should be provided.

We agree with the Reviewer that in this cell cycle phase AurkA increased nuclear localization is paralleled by an overall increase in AurkA levels, so that is not easy to define a real nuclear “enrichment”. We have therefore quantified both AurkA nuclear

levels and the N/C ratio in these cells and shown the data in new Fig. 1E. We have also modified the text to take into account this observation and more accurately describe the observed phenomenon.

- Fig. 2B: the authors describe that "the percentage of AurkA positive cells decreased after LMB treatment". This does not correspond to the micrograph shown, where we see a global increase of AURKA-positive signal in cells compared to the control condition and regardless of the localization of the kinase. How do authors explain that? This is somehow puzzling when looking at the blot in fig. 2C, where we see a loss of AURKA upon LMB treatment. Could it be that a specific fraction of AURKA - that remains visible on the image - is lost in western blotting analyses?

From the reviewer comment, we realized that the panel we had chosen for Figure 2B was misleading. We do observe a decrease of AurkA-positive cells, as assessed in the IF quantification and WB analyses. We have replaced the panel and hope the images now more clearly represent the situation depicted in the graphs and WB.

- MG132 is a non-specific inhibitor of the proteasome, and under some conditions it is known to trigger proteasome-independent effects such as autophagy induction. Can the authors corroborate their analyses with a more specific inhibitor such as epoxomicin? Plus, could the MG132 effect on the number of cells with nuclear AURKA be induced by its indirect effect on the cell cycle rather than a specific effect on AURKA? What is the cell cycle profile of MG132-treated cells in the experiments performed in this manuscript?

We have repeated the analyses shown in Figure 2B and 4D using Epoxomicin. Data support the conclusions from the MG132 analyses and are shown in Supplementary Figures 1B and 2C, where we also show (Supplementary Figure 1C) the FACS profiles of MG132 and Epoxomicin treated hTERT RPE1 cells. Quantification of the G2/M fraction from triplicate experiments indicate a mild, if any, increase under MG132 and Epoxomicin treatments (25-26% compared to 21-22% in control cultures), that cannot account for the significant fraction (about 80%) of cells displaying nuclear AurkA under these conditions.

- The authors state the fact that protein degradation is an important negative regulator of overexpressed AURKA, but they do not exclude that it is the same when the kinase is expressed at endogenous levels. Basically, are the mechanisms the same in the two conditions or not? In a cancer-related perspective, I believe it would be important to differentiate between these two conditions by assessing the behavior of the kinase with same parameters overall.

We agree with the reviewer that investigating the regulation of the levels of endogenous AurkA in interphase is an interesting issue. Still, no direct comparison can be applied, since endogenous AurkA is subjected to a strict cell cycle dependent regulation at the transcriptional level. When we add MG132 to an asynchronous culture of hTERT RPE1, we do see a slight increase of cells displaying nuclear AurkA (confidential information, see histogram below; data obtained from hTERT-RPE 1 cultures treated with MG132 or Epoxomicin using the same protocols described for Figures 2B and 4D, and Supplementary Figures 1B and 2C, and stained with AurkA antibodies in IF experiments). Since we also see a slight increase of G2/M cells by FACS analyses (see new data in Supplementary Figure 1C) we do not feel comfortable in showing this piece of data without a more extensive characterization which would take into account the cell cycle phases and the mRNA levels of AurkA.

Also, we consider that an interesting possibility is that regulation of AurkA at the protein level, downstream of transcriptional activation, may be involved in the recently emerging interphasic roles reported for the kinase, that may be subjected to modulation following stimuli, such as DNA damage or replication stress. We plan to investigate these possibilities in future studies.

- Fig. 4B: please provide quantifications of PLA spots.

We have quantified isPLA signals using two distinct analyses (% of cells displaying nuclear isPLA signals and isPLA signal intensity in nuclei) and extended the analysis to the overexpressing cell lines (see next point). New data are shown in Fig. 4B.

- Can the authors explain why cells co-expressing AURKA and TPX2 show no differences in the % of cells positive for AURKA upon LMB treatment? Could this be due to alterations in the cell cycle profile compared to AURKA-only overexpressing cells? Is this due to the direct interaction of AURKA and TPX2 in the nucleus which becomes stronger upon co-expression, thereby retaining AURKA in the nucleus? Similarly, can AURKA S155R still interact with TPX2, and conversely, can TPX2^{Δ43} still interact with AURKA?

As the reviewer points out, we believe that TPX2 is the limiting factor for the nuclear accumulation of AurkA. Indeed, the fraction of nuclear-localized AurkA interphases is already higher in TPX2 co-overexpressing cells in respect to AurkA overexpressing cells treated with LMB. As the reviewer points out, this mechanism would rely on an increased interaction of AurkA and TPX2 upon co-expression. To demonstrate this, we have now performed isPLA assays in the overexpressing cells lines too. Images and quantifications are shown in new Figure 4B and indicate that the nuclear interaction is stronger in the AurkA/TPX2 OE cell line, compared to control and AurkA OE cells. Concerning the mechanisms, we cannot rule out that, besides protecting AurkA from proteasome-dependent degradation, TPX2 also competes with the export machinery. This will be an interesting hypothesis to investigate in the future as suggested in the Discussion section.

Finally, concerning TPX2 mutants, the 1-43 region is well characterized in the literature for being responsible for AurkA binding (Bayliss et al., 2003) and mutants/truncations in this region are widely used as TPX2 versions defective for AurkA interaction (see for example Tasi and Zheng, 2005; Bird and Hyman, 2008; Sardon et al., 2008; Giubettini et al., 2011; Polverino et al., 2021). As for the S155R, we have recently

shown (Abdelbaki et al., 2022, LSA) that it displays reduced, although not abolished, interaction with TPX2, consistent with data in the literature (Bibby et al., 2009). We have added reference to the literature in the text.

- Fig. 5B: There is an increased abundance of TPX2 in AURKA-only overexpressing cells. Can the authors comment on that? It could be that there is a feedback loop to increase TPX2 stability under these conditions, but that since there is no difference at the mRNA level, TPX2 becomes the limiting factor to achieve a full expansion of mammospheres (such as the one seen upon co-expression).

The reviewer noticed a slight increase in TPX2 levels in the AurkA OE cell line. By quantification, this is an about 1.5 fold increase. Indeed TPX2 has been recently reported as a Myc target gene (Rohrberg et al., 2020) and Myc transcription is positively regulated by nuclear AurkA (Zheng et al., 2016). Therefore, the reviewer's hypothesis is intriguing and certainly deserves attention in future investigations. Still, at the current stage we feel we do not have enough data to support it and therefore we did not add a specific comment on it in the text.

- I somehow disagree with the following statement "these data support the idea that overexpressed TPX2, by increasing the levels of nuclear overexpressed AurkA, plays a key role in mediating the oncogenic functions described for nuclear AurkA". The authors do not rule out the possibility that TPX2 is not needed to increase AURKA levels, but to fully activate it instead. This is also visible on the blot in Fig. 5B, where the co-expression seems to slightly lower AURKA levels instead of increasing its abundance. Either they demonstrate that upon TPX2 co-expression, overexpressed nuclear AURKA needs TPX2 to get fully active, either I would suggest rephrasing this conclusion.

Most previously characterized AurkA nuclear oncogenic functions to which we refer to in Figure 5 are reported to be kinase-independent (Zheng et al., 2016; Naso et al., 2021). For this reason, we did not focus our attention on the activating role of TPX2 on AurkA for this specific issue. In replying to Reviewer 1 we have now also added information showing that the kinase activity of AurkA is not required for its nuclear accumulation. Finally, we have not noticed signals for AurkA pThr288 in interphase nuclei in the different cell lines, suggesting that interphase activation-if any- may be subjected to a more complex regulation than just TPX2 binding. Still, we agree that we did not formally demonstrate under our conditions that the effects of AurkA/TPX2 co-OE in Figure 5 are not due to increased kinase activity. Thus, we rephrased the text to take into account the reviewer's comment and the above mentioned considerations.

- I just wanted to point out that fluorescent micrographs are sometimes of bad quality (e.g. Fig. 1A, 2A, 2B). However, this could be due to the PDF conversion.

We do not notice significant differences from the other panels in our images 1A-2A-2B, which are all acquired and elaborated using comparable methods. Following a previous comment, panel 2B has nonetheless been modified, and hopefully images appear of good quality. Should panels in Figure 1A and 2A look not of sufficient quality we will be happy to provide them in different file formats.

January 27, 2023

RE: Life Science Alliance Manuscript #LSA-2022-01726-TR

Dr. Giulia Guarguaglini
National Research Council
Laboratories of genetics
Via degli Apuli 4
Via degli Apuli, 4
Rome 00185
Italy

Dear Dr. Guarguaglini,

Thank you for submitting your revised manuscript entitled "AurkA nuclear localization is promoted by TPX2 and counteracted by protein degradation". We would be happy to publish your paper in Life Science Alliance pending final revisions necessary to meet our formatting guidelines.

- please upload your manuscript text as an editable doc file
- please upload your main and supplementary figures as single files
- please upload your table files as editable doc or excel files
- please add ORCID ID for secondary corresponding author-they should have received instructions on how to do so
- please add the Twitter handle of your host institute/organization as well as your own or/and one of the authors in our system

A. FINAL FILES:

B. MANUSCRIPT ORGANIZATION AND FORMATTING:

Sincerely,

Reviewer #1 (Comments to the Authors (Required)):

I thank the authors for answering my two questions in their rebuttal and based on those minor changes, the manuscript warrants publication

Reviewer #2 (Comments to the Authors (Required)):

I would like to thank the authors for taking the time to answer my comments and for better explaining their rationale. The new experiments/additions to the text look convincing and further corroborate the study and its main results. I also believe that the confidential data provided participate in reinforcing the main conclusions of the study. Thank you for sharing them. I have no further questions for the authors, and I'll be happy to see the study online.

February 1, 2023

RE: Life Science Alliance Manuscript #LSA-2022-01726-TRR

Dr. Giulia Guarguaglini
National Research Council
Laboratories of genetics
Via degli Apuli 4
Rome 00185
Italy

Dear Dr. Guarguaglini,

Thank you for submitting your Research Article entitled "AurKA nuclear localization is promoted by TPX2 and counteracted by protein degradation". It is a pleasure to let you know that your manuscript is now accepted for publication in Life Science Alliance. Congratulations on this interesting work.

DISTRIBUTION OF MATERIALS:

Again, congratulations on a very nice paper. I hope you found the review process to be constructive and are pleased with how the manuscript was handled editorially. We look forward to future exciting submissions from your lab.

Sincerely,
